# Brief Communication: On the extremeness of the July 2021 precipitation event in western Germany

Katharina Lengfeld[1], Paul Voit[2], Frank Kaspar[1], and Maik Heistermann[2]

[1]Deutscher Wetterdienst (DWD), Offenbach, Germany
[2]Institute for Environmental Sciences and Geography, University of Potsdam, Potsdam, Germany

**Correspondence:** Katharina Lengfeld (katharina.lengfeld@dwd.de)

**Abstract.**

The weather extremity index (WEI) and the cross-scale WEI (xWEI) are useful to determine the extremeness of precipitation events. Both require the estimation of return periods across different precipitation duration levels. For this purpose, previous studies determined annual precipitation maxima from radar composites in Germany, and estimated the parameters of a generalized extreme value distribution (GEV). Including the year 2021 in the estimation of GEV-parameters, the devastating event in July 2021 drops from first to fourth rank regarding the WEI compared to all events between 2001 and 2020, but remains the most extreme regarding the xWEI. This emphasizes that it was extreme across multiple spatial and temporal scales, and the importance of considering different scales to determine the extremeness of rainfall events.

## 1 Introduction

In July 2021, an extreme precipitation event took place in western Germany (Fig. 1 left) and neighbouring countries which caused one of the most severe natural disasters in Germany and Europe. In Germany, more than 180 people lost their lives (more than 140 in Rhineland-Palatinate, and more than 40 in North Rhine-Westphalia). According to Munich Re, loss and damage amounted to EUR 46 billion (MunichRE, 2022), 33 billion in Germany alone. The German Insurance Association (GDV) reported a new record of EUR 8.2 billion insured flood losses for a single event (GDV, 2021). Numerous studies have been published since the event, e.g. related to meteorological, hydrological, and impact-related aspects (Mohr et al., 2023), hydro-geomorphological processes (Dietze et al., 2022), or early warning (Fekete and Sandholz, 2021; Thieken et al., 2022a). The temporal development of the event (Fig. 1, right) indicates heavy precipitation at several scales from showers of a few hours up to continuous rainfall for more than a day. With regard to the extremeness of the event's precipitation, Germany's national meteorological service (Deutscher Wetterdienst; DWD hereafter) estimated return periods of well over 100 years for large parts in North Rhine-Westphalia and Rhineland-Palatinate (Junghänel et al., 2021). Mohr et al. (2023) reported return periods of more than 700 years over large areas.

Junghänel et al. (2021) and Dietze et al. (2022) highlighted that an important feature of the July 2021 precipitation event was not only high rainfall accumulation at a large spatial extent, but also the occurrence of high return periods at various durations as well as a remarkable heterogeneity in space. In order to account for such complex events, and to formalize the quantification

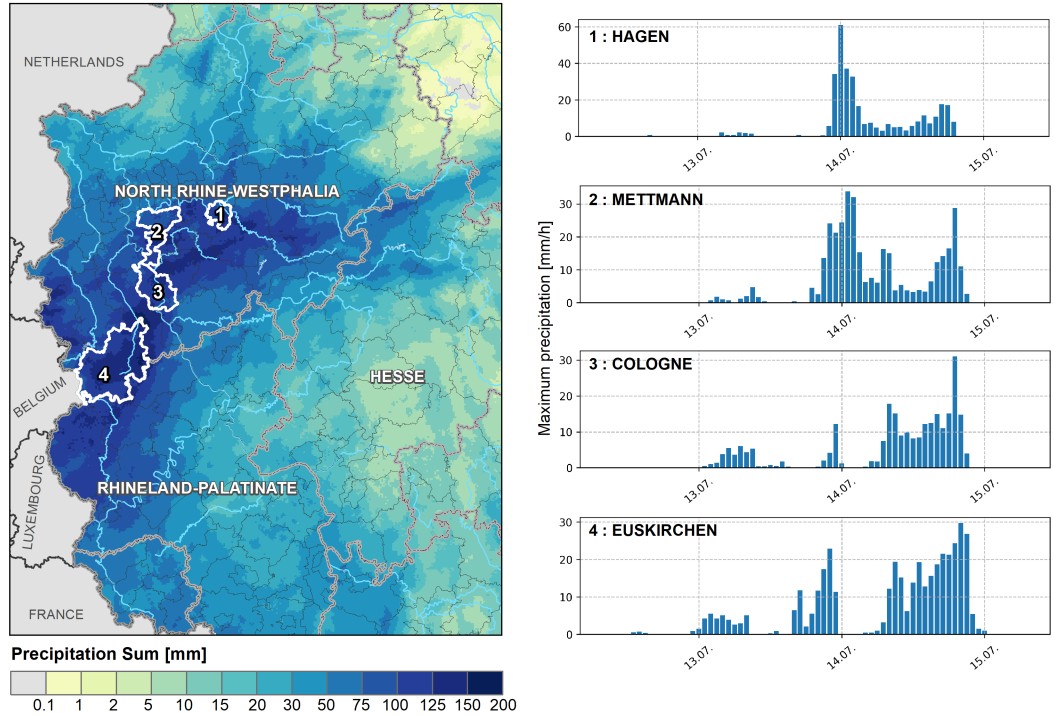

**Figure 1.** Left: Accumulated rainfall of the precipitation event from 12. July, 5:50 UTC to 15.July 2021, 5:50 UTC based on RADKLIM v2017.002 (Winterrath et al., 2018b). Right: Temporal development of the maximum hourly precipitation within the four districts Hagen, Mettmann, Cologne and Euskirchen.

of their extremeness, Müller and Kaspar (2014) had suggested (in this journal) the weather extremity index (WEI). The WEI identifies the spatial and temporal scale at which an event was most extreme and allows to quantify the extremeness (rarity) of an event by averaging return periods of all affected pixels and multiplying it with a measure of the spatial extent of this event (see Section 3). It has been increasingly used since then (Gvoždíková et al., 2019; Minářová et al., 2018), and was also established as a routine measure of extremeness by the DWD (Lengfeld et al., 2021). Based on the German operational radar data, DWD had quantified the WEI of the event to be 227 ln(yr)km. With this value, the event had outranked all other events based on radar data that were previously classified in the period from 2001 to 2020.

Recently, Voit and Heistermann (2022) argued in their contribution to this special issue that the WEI could be supplemented in order to account for events that are not only extreme at a distinct spatial and temporal scale. Instead, such events could be extreme across various scales - a feature that also appeared to characterize the July 2021 event. Hence, Voit and Heistermann defined the cross-scale weather extremity index (xWEI), and found that, on the basis of xWEI, the July 2021 event outranked other extreme events by far.

However, the estimates of WEI and xWEI that have so far been obtained for the July 2021 event by the DWD and Voit and Heistermann (2022) share a limitation: the parameters of the underlying generalized extreme value (GEV) distributions

were estimated based on the RADKLIM data set - a reanalysis of DWD's procedure for radar-based quantitative precipitation estimation. As the RADKLIM data used for that purpose did not yet contain data from 2021, the July 2021 event was not included in the estimation of GEV parameters. The extremeness of the event might hence have been overestimated.

In June 2022 the DWD published the updated RADKLIM reanalysis (Winterrath et al., 2018b) which now includes data from 2001 to 2021. This provides the opportunity to re-assess, in this brief communication, the extremeness of the July 2021 event on the basis of the most recent homogeneous radar reanalysis, using both the established WEI and the xWEI as a supplement to account for extremity across scales.

In section 2, we introduce the underlying datasets, RADOLAN, RADKLIM and CatRaRE. Section 3 summarizes the concept and the computation of the two extremity indices, WEI and xWEI. In section 4, we present an assessment of the five highest ranking extreme events in the most recent RADKLIM dataset, and the changes introduced by including or excluding the data from the year of their occurrence. Section 5 summarizes our findings and highlights implications for research and risk management.

## 2    Data

Our analysis of the July 2021 event is based on two different sets of gauge-adjusted, hourly precipitation data: (i) The operational RADOLAN-RW product (RADOLAN, 2022) always uses the most recent algorithms for data processing and is available in realtime; (ii) RADKLIM (RADKLIM v2017.002, see Winterrath et al., 2018a, b), in turn, is a reanalysis of radar data since 2001: it is based on a homogeneous set of algorithms, includes advanced climatological corrections, and uses additional hourly and daily rainfall data from rain gauges that were not available in realtime.

Based on RADKLIM, DWD provides a Catalogue of Radar-based heavy Rainfall Events (CatRaRE) containing rainfall events in Germany of 11 durations between 1 and 72 hours since 2001 that exceed DWD's warning level 3 for severe weather in terms of precipitation rate. A detailed description of the catalogue and its parameters can be found in Lengfeld et al. (2021), the data can be accessed via Lengfeld et al. (2022).

## 3    Methods

Müller and Kaspar (2014) defined the so-called weather extremity index WEI that takes into account the affected area $A$ and the rareness of an event (in form of the return period $T$ [yrs]) for various durations. At a given duration $t$ [h], the extremity $E_{t,A}$ [ln(yr)km] amounts to:

$$E_{t,A} = \frac{\sum_{i=1}^{n} ln(T_{t,i})}{A} \cdot \frac{\sqrt{A}}{\sqrt{\pi}}, \tag{1}$$

for grid points i. The maximum $E_{t,A}$ over all durations is then defined as WEI and determines the spatial extent and duration of the most extreme stage of an event.

Following Lengfeld et al. (2021), we used DWD's national radar composite data to first determine annual precipitation maxima, and then estimate the GEV parameters. The whole area affected by the event, as defined in CatRaRE, is taken into

account for the calculation of the WEI. We used two different combinations of radar data and GEV parameter estimates: (1) The realtime setup with estimation of GEV parameters based on RADKLIM data from 2001 to 2020 (not including the July 2021 event) combined with operational hourly rainfall sums from RADOLAN and (2) the climatological setup with estimation of GEV parameters based on RADKLIM data from 2001 to 2021 combined with reprocessed hourly rainfall sums from RAD-KLIM. The dataset only spans 21 years of observations. Therefore, we assumed stationarity within the dataset. The differences between the RADOLAN and the RADKLIM data did not influence the results significantly (refer to Tab. 1). Therefore, the focus of this study lies on the effect of including or excluding data from 2021 in the estimation of GEV parameters. In order to contextualize this effect, we estimated the GEV parameters and hence the WEI for the five highest ranking events between 2001 and 2021 with and without including their year of occurrence.

The retrieval of xWEI was documented in detail by Voit and Heistermann (2022). The fundamental idea of xWEI is to integrate the extremeness $E_{tA}$ across spatial scales and all regarded duration levels instead of searching for a maximum of $E_{tA}$. This integration is, again, based on the aforementioned $E_{tA}$-curves which, for each duration, display $E_{tA}$ across spatial scales. We now interpret these curves in a 3-dimensional coordinate system in which the x-axis represents the area, the y-axis the duration, and the z-axis $E_{tA}$. Together, these curves span a surface. The volume underneath this surface corresponds to the cross-scale extremeness (xWEI) of an event (Voit and Heistermann, 2022). To keep the results comparable we used the same bounds for the computation of the xWEI as in our previous study (Voit and Heistermann, 2022): Duration levels from 1 to 72 hours and a bounding box of 200 km * 200 km around the centroid of the event:

$$xWEI = \int\limits_{ln(t=1h)}^{ln(72h)} \int\limits_{A=1km^2}^{(200km)^2} E_{tA}\, dA\, d(ln(t)) \tag{2}$$

## 4 Results

Figure 2 illustrates how the $E_{tA}$-curves changed after the most recent RADKLIM reanalysis was included in the estimation of GEV parameters in the "climatological" setup: as expected, $E_{tA}$ reaches lower values across all durations and areas than in the "realtime" setup. On average, maximum $E_{tA}$-values are 20-30 % lower for all relevant durations. The overall structure of the $E_{tA}$-curves remains dominated by long durations between 12 and 72 hours. Shorter durations of 4 to 6 hours do not reach $E_{tA}$ values of 100 ln(yr)km, durations up to three hours are even less pronounced. The most extreme stage for the July event is in both cases reached for a duration of 48 hours.

Tab. 1 puts WEI and xWEI of the five most extreme events according to CatRaRE v2022.01 into context, and also shows the changes of WEI and xWEI for the events, depending on including or excluding the year of their occurrence for quantification. For the July 2021 event, the values of WEI and xWEI changed from 227 to 179 ln(yr)km (-48), and from 3731 to 3134 (-597), respectively. Based on the updated WEI values, the July 2021 event no longer outranks all other events, but now ranks at the 4th position. The most extreme event according to the WEI with a value of 208 ln(yr)km occurred in Saxony in August 2002 and led to the devastating flood along the Elbe river. This event had the largest extremity at a duration of 24 hours and

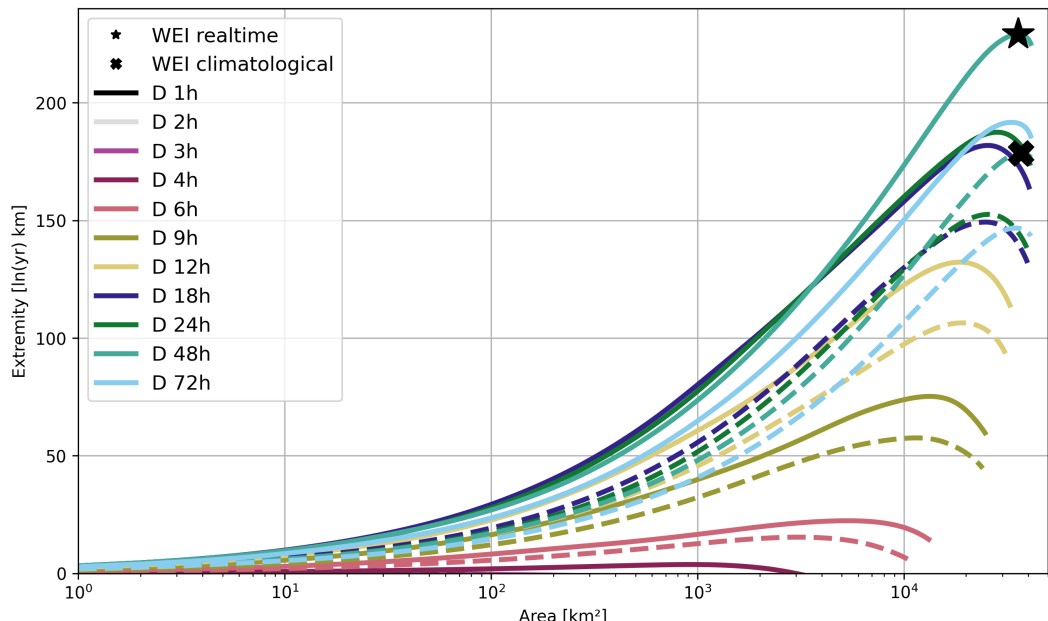

**Figure 2.** Extremity calculated for 14 July 2021, 23:50 UTC, with operational RADOLAN data and GEV parameters obtained from RAD-KLIM 2001-2020 (realtime setup: solid lines, without including the year 2021 in the GEV parameter estimation, WEI marked with a star) and from reprocessed RADKLIM data (Version 2017.002) and GEV parameters from RADKLIM 2001-2021 (climatological setup: including the year 2021 in the GEV parameter estimation, dashed lines, WEI marked with an X)

precipitation of 87 mm in these 24 hours averaged over the affected area. The events in second and third place in July 2017 and 2002 both lasted 48 hours and are, thus, more similar to the July 2021 event. Their WEI is only slightly higher with 188 ln(yr)km and 180 ln(yr)km, respectively. Both events have lower mean precipitation values of 85 mm compared to the July 2021 event with 91 mm, but larger spatial extents. This underlines the sensitivity of the WEI to the affected area. Note that

due to computational efficiency only a fixed set of duration levels (1, 2, 3, 4, 6, 9, 12, 18, 24, 48 and 72 h) was considered. The actual duration at which the events were most extreme might be somewhere between these fixed levels. With regard to the updated xWEI, however, the July 2021 event (still) appears as the most extreme event out of these five. If ranked according to xWEI, the Saxony 2002 event would be on second rank, followed by the Berlin 2017 event and the two events in Lower Saxony (2002 and 2017).

WEI and xWEI of the other four events in Tab. 1 did not change with the updated GEV parameters when excluding the year 2021 in their estimation. However, we repeated the computation of WEI and xWEI by leaving out the year of the events' occurrences in the estimation of the GEV parameters (values in parentheses). That way, we could appraise the sensitivity of WEI and xWEI to the underlying database. In all cases, both WEI and xWEI increased noticeably by 35 to 48 ln(yr)km and 482 to 795, respectively, when the year of occurrence was excluded from the GEV parameter estimation. Yet, considering only

the values in parenthesis, the ranking remained the same for both WEI and xWEI.

**Table 1.** The five most extreme precipitation events in CatRaRE v2022.01, according to the weather extremity index (WEI), and the corresponding cross-scale weather extremity (xWEI) values for the same events. In parenthesis, we show the values obtained with GEV parameters that do not account for the year of the event's occurrence. For the July 2021 event, the second value in parenthesis was obtained with the realtime setup (based on operational radar data and without including the year 2021 in the GEV parameter estimation). Duration and area specify the duration and area at which the event reached its maximum extremity, $\overline{R}$ is the areal average of the event's rainfall depth.

| Rank | Region | Date | ID | Duration [h] | Area [km$^2$] | $\overline{R}$ [mm] | WEI [ln(yr) km] | xWEI [-] |
|------|--------|------|-----|-------------|--------------|--------------------|-----------------|----------|
| 1. | Saxony | August 12-13, 2002 | 1798 | 24 | 48420 | 87 | 208 (243) | 2855 (3650) |
| 2. | Lower Saxony | July 24-26, 2017 | 17961 | 48 | 54287 | 85 | 188 (235) | 2121 (2603) |
| 3. | Lower Saxony | July 17-19, 2002 | 1239 | 48 | 45053 | 85 | 180 (228) | 2402 (2919) |
| 4. | West-Germany | July 13-15, 2021 | 24193 | 48 | 41177 | 91 | 179 (227, 232) | 3134 (3731) |
| 5. | Berlin/Brandenburg | June 29-30, 2017 | 17695 | 24 | 33927 | 72 | 165 (208) | 2577 (3137) |

## 5 Conclusions

Quantifying the extremeness of precipitation events and ranking past events accordingly can help researchers, stakeholders and the general public to objectively put different extreme events into a context. Based on the weather extremity index (WEI) and the cross-scale weather extremity index (xWEI), we re-assessed the extremeness of the disastrous heavy rainfall event which took place in western Germany in July 2021. To that end, we used the most recent reanalysis of DWD's climatological weather radar data (RADKLIM).

While the impact of the July 2021 event was unique, our analysis reveals that it was just the fourth most extreme event in the period from 2001-2021, according to the WEI. Before having included the recent RADKLIM data in the GEV parameter estimation, the event had been considered the most extreme. This shows how sensitive WEI and xWEI are to the underlying extreme value statistics, especially because we are dealing with quite short radar time series (20 years). According to the xWEI, however, we found that the July 2021 event still outranks all other events within the recent RADKLIM dataset. This emphasizes, on a formal basis, previous reports that one of the key features of the July 2021 event was its extremeness across spatial and temporal scales. Thieken et al. (2022b) framed the Elbe flood event that followed the August 2002 precipitation (rank 1 in Tab. 1) as a "compound inland flood", in the sense that different flood mechanisms overlayed and amplified each other. Similar observations were made for the July 2021 event (e.g. Mohr et al. (2023)). Given the exceptional xWEI values of both the August 2002 and the July 2021 events, it might be worth to discuss, in the future, whether heavy precipitation events that are extreme across scales could be framed as hydro-meteorological compound events. Mohr et al. (2023) also suggested that, originating from the weeks before the event, the antecedent soil moisture in the affected region was relatively high, favoring the occurrence of saturation excess and thus an enhanced rainfall-runoff transformation.

But while the xWEI implies that the July 2021 event in fact outranked others, we need to acknowledge that the top ranking events are relatively close in terms of WEI and xWEI. Hence, the resulting rankings should not be over-interpreted, especially if we consider uncertainties regarding the radar rainfall estimation, the estimation of return periods and the selection of the spatial domain. Indeed, the highest ranking events appear quite similar with regard to their extremeness. Their occurrence in very different regions of Germany suggests that such types of events could effectively take place anywhere in Germany, and might be considered as hydro-meteorological reference events for research and disaster risk management throughout the country. The *impact* of such events on the ground, however, will vary dramatically based on governing hydrological and hydro-geomorphological processes as well as exposure and vulnerability. This is demonstrated by the two extreme events that occurred in Lower Saxony, the impacts of which appeared to be less severe, or at least were less visible in the national media. We hence consider it important to assess the potential impacts of such events anywhere across Germany or Central Europe.

*Code and data availability.* The RADKLIM dataset is publicly available at the DWD open data servers: https://dx.doi.org/10.5676/DWD/RADKLIM_RW (Winterrath et al., 2018b) as well as the CatRaRE catalog (https://dx.doi.org/10.5676/DWD/CatRaRE_W3_Eta_v2022.01) (Lengfeld et al., 2022). The code and exemplary data for the computation of xWEI is published in following repository: doi.org/10.5281/zenodo.6556463 (Voit, 2022).

*Author contributions.* All authors conceptualized the study, KL and PV carried out the statistical analysis, KL prepared the figures, all authors prepared the manuscript.

*Competing interests.* The contact author has declared that neither of the authors has any competing interests.

*Acknowledgements.* Parts of this research was supported by the Deutsche Forschungsgemeinschaft (grant no. GRK 2043, project number 251036843).

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
