# Peer review of "Brief Communication: On the extremeness of the July 2021 precipitation event in western Germany"

_EGUsphere, 2022_

## Author Comment (AC1)

**Interactive Discussion: Author Response to Referee #1**

**Brief Communication: On the extremeness of the July 2021 precipitation event in western Germany**

Lengfeld et al.
*EGUShpere,* `doi:10.5194/egusphere-2022-979`
* * *
**RC:** *Reviewer Comment*,    AR: *Author Response*,    ☐ Manuscript text

Dear Referee,

we would like to thank you very much for your willingness to review this paper, and for your swift, positive and constructive response to the manuscript.

Please find our responses to your comments below. These should be considered as preliminary (part of the interactive discussion). The final implementation of changes also depends on another referee report.

Thanks again for your efforts!

Kind regards,
Katharina Lengfeld (on behalf of the author team)

**RC:** *[...] L3: firstly, the sentence "both rely....GEV parameters" might not be clear to everybody (I am putting myself in the shoes of someone not completely within the topic). Secondly, from how it is written it seems that the GEV distribution is the only one that can be used for the estimation of return periods of extreme events, which is not the case. I would suggest trying to make this sentence clearer and include some details about the link between return period and GEV parameters when you introduce the WEI in the main text.*

AR: We agree that this sentence should be clearer, and it is true that other data or methods could be used to estimate the required return periods. Yet, we should not go too much into the computational details within the abstract. We have tried to rephrase along these requirements.

> [...] Both require the estimation of return periods across different precipitation duration levels. For this purpose, previous studies have used national composite radar data in Germany to determine annual precipitation maxima, and used this to estimate the parameters of a generalized extreme value distribution (GEV). When including the year 2021 [...]

**RC:** *L26: even if the authors claim that the WEI is increasingly used in the community (I guess they mean the meteorological one), for someone that is not within it the information reported about the WEI (229 log(yr)km) at this point of the paper is not straightforward to understand. The definition of WEI is indeed provided only at L60. I would therefore suggest trying to insert this number into a context and explain briefly what the extremity index is and how it is computed (or at least what variables are considered) such that a broader audience can have a feeling of what this number*

AR: The unit of the WEI is indeed hard to communicate which is why we described the general concept underlying the WEI starting in line 22 of the preprint:

*"The WEI identifies the spatial and temporal scale at which an event was most extreme and allows to quantify the extremeness (rarity) of an event."*

To put the mentioned value of 229 log(yr)km into context the following sentence in the preprint states:

*"With this value, the event outranked all other events based on radar data that were previously classified in the period from 2001 to 2020."*

To further clarify this, we will add a cross link to section 3 and the following sentence in line 24:

> The WEI identifies the spatial and temporal scale at which an event was most extreme and allows to quantify the extremeness (rarity) of an event by averaging return periods of all affected pixels and multiplying it with a measure of the spatial extent of this event (see Section 3).

We hope that this is an adequate compromise between summary and detail, given that the "Brief Communication" format provides limited space.

RC: *L57-66: see my comments on L3.*

AR: We will replace, in l. 63 of the preprint, the sentence "We use two different combinations of radar data and GEV parameter estimates" by the following statement in order to make clear that our choice to use DWD radar data together with the GEV distribution builds on previous research in that direction. However, we do not find it useful for the reader, at this point of the manuscript, to go into detail about alternative data or methods in order to retrieve the required return periods.

> Following Lengfeld et al. (2021), we used DWD's national radar composite data to first determine annual precipitation maxima, and then estimate the GEV parameters.

RC: *L69-74: writing explicitly the formula through which the xWEI is computed would be helpful.*

AR: Practically the xWEI is computed by an interpolation of a surface over Eta-curves Formally this corresponds to a double integral which can be expressed by following formula, which we will add in line 74:

$$xWEI = \int_{ln(t)} \int_A E_{tA} \, dA \, d(ln(t)) \tag{1}$$

RC: *Figures are not color-blind friendly, please consider change color scaled such that everyone can appreciate differences and color meanings.*

AR: We prepared Figure 1 and 2 with colorblind-friendly colormaps (see figures below), and we will use these in the manuscript. We used YlGnBu from ColorBrewer 2.0 (https://colorbrewer2.org/type=sequentialscheme=YlGnBun=9) for the colormap in Figure 1 and Tol_muted (https://zenodo.org/record/3381072.Y5MZ0iUxlhF) for the colors in Figure 2.

[Figure]

**RC:** *Figures 2: do the gray lines represent the WEI of the July 2021 event? If yes, it should be specified somewhere. Moreover, I would rather representing it as a point (with different markers/colors depending on the RADKLIM product used to compute it).*

AR: Yes, the grey lines represent the WEI of the July 2021 event for the two different setups. We will change the lines to symbols and mark the WEI of the realtime setup with a star and of the climatological setup with an X in the new version of Figure 2 as shown in the figure below. We will also add the information to the caption of Figure 2.

[Figure]

Figure 1: Extremity calculated for 14 July 2021, 23:50 UTC, with operational RADOLAN data and GEV parameters obtained from RADKLIM 2001-2020 (realtime setup: solid lines, WEI marked with a star) and from reprocessed RADKLIM data (Version 2017.002) and GEV parameters from RADKLIM 2001-2021 (climatological setup: dashed lines, WEI marked with an X)

---

## Author Comment (AC3)

**Interactive Discussion: Author Response to Referee #3**

**Brief Communication: On the extremeness of the July 2021 precipitation event in western Germany**

Lengfeld et al.

*EGUShpere,* `doi:10.5194/egusphere-2022-979`
* * *
**RC:** *Reviewer Comment*,     AR: *Author Response*,     ☐ Manuscript text

Dear Referee,

we would like to thank you very much for your willingness to review this paper, and for your swift, positive and constructive response to the manuscript.

Please find our responses to your comments below. These should be considered as preliminary (part of the interactive discussion). The final implementation of changes also depends on another referee report.

Thanks again for your efforts!

Kind regards,
Katharina Lengfeld (on behalf of the author team)

**RC:** *[...] The two WEI values are not obvious in Fig. 2. Moreover, they are labeled as "realtime" and "climatological" which is not the main factor of the difference as authors state at l. 66-67.*

AR: We will highlight the two WEI values with two different symbols instead of the grey lines (see new figure below) to make the difference clearer. The reviewer is right, the difference is mainly caused by including/excluding the year 2021 in the estimation of the GEV parameters. For the sake of simplicity, and to improve the readability of the manuscript, we had chosen the terms "realtime" and "climatological" for the two different scenarios as defined in lines 63-66, and hence use these names for the labels in Figure 2. To emphasize the difference between the two setups again in the figure, we will also mention it in the figure caption and change it as follows:

> Extremity calculated for 14 July 2021, 23:50 UTC, with operational RADOLAN data and GEV parameters obtained from RADKLIM 2001-2020 (realtime setup: solid lines, without including the year 2021 in the GEV parameter estimation, WEI marked with a star) and from reprocessed RADKLIM data (Version 2017.002) and GEV parameters from RADKLIM 2001-2021 (climatological setup: including the year 2021 in the GEV parameter estimation, dashed lines, WEI marked with an X).

**RC:** *Significant decrease in both WEI and xWEI values when including the year 2021 proofs that the presented estimation of return periods is not very robust. It is natural because of the rather short length of the data series. Nevertheless, because this is the main concern of the communication, a short sensitivity analysis could be done, e.g. by the "leave-one-out" technique.*

[Figure]

AR: We agree that an analysis in terms of the suggested leave-one-out technique would be a useful extension in order to assess the sensitivity of the indices to the fact whether an event is included in the GEV parameter estimation or not. For the revised manuscript, we will calculate WEI and xWEI for the other four events in Tab. 1 using GEV parameters computed *without* the years of their occurrence, and add this information to Table 1.

RC: *The unique impacts of July 2021 were not only because of the precipitation event itself but partly also due to the enhanced soil moisture at its beginning. In my opinion, this fact should be more stressed in the discussion.*

AR: We agree with the reviewer that the antecedent soil moisture might have had an effect on runoff formation. We would prefer, however, to be a bit cautious with regard to that effect: first, the antecedent wetness is not directly related to the event-based extremity indices of precipitation; second, the role of antecedent wetness with regard to runoff formation and hence the impacts of the events is still subject to scientific discussion. On that basis, we suggest to add the following statement to the conclusions section, after l. 103 of the original manuscript:

> Mohr et al. (2022) also suggested that, originating from the weeks before the event, the antecedent soil moisture in the affected region was relatively high, favoring the occurrence of saturation excess and thus an enhanced rainfall-runoff transformation.

**RC:** *line 80: adding the word "even" into the sentence ("are even less pronounced") would make it more clear in my opinion;*

 AR:  Thanks for the comment. We will add the word "even" to the sentence.

**RC:** *line 84: I suggest to add "respectively" at the end of the sentence.*

 AR:  We will add "respectively" at the end of the sentence.

**References**

Mohr, S., Ehret, U., Kunz, M., Ludwig, P., Caldas-Alvarez, A., Daniell, J. E., Ehmele, F., Feldmann, H., Franca, M. J., Gattke, C., Hundhausen, M., Knippertz, P., Küpfer, K., Mühr, B., Pinto, J. G., Quinting, J., Schäfer, A. M., Scheibel, M., Seidel, F., and Wisotzky, C.: A multi-disciplinary analysis of the exceptional flood event of July 2021 in central Europe. Part 1: Event description and analysis, Nat. Hazards Earth Syst. Sci. Discuss. [preprint], https://doi.org/10.5194/nhess-2022-137, 2022.

---

## Author Response (AR1)

**Interactive Discussion: Author Response to Editor**

**Brief Communication: On the extremeness of the July 2021 precipitation event in western Germany**

Lengfeld et al.
*EGUShpere,* `doi:10.5194/egusphere-2022-979`
* * *
**EC:** *Editor Comment*,    **RC:** *Referee Comment*,    **AR:** *Author Response*,    ☐ Manuscript text

Dear Prof. Ulbrich, dear referees,

we would like to thank all of you, again, for your thoughtful comments. Please find our responses to the referees (which correspond to the initeractive discusssion) and to the Editor's comments below.

Thanks again for your efforts!

Kind regards,
Katharina Lengfeld (on behalf of the author team)

**1.  Response to the Editor's comments**

**EC:**  *Formula for the xWEI (equation 1). To my understanding, the xWEI is computed referring to a maximum value in terms of a measure combining temporal and spatial extension of the event. Thus, I would expect defined integral bounds in the formula. Can you please clarify, and, if I am right, include the bounds?*

**AR:**  The xWEI is - in contrast to the WEI - not referring to a maximum value of a measure, but it integrates Eta over spatial extent and duration. In that sense, the editor is correct in stating that xWEI has defined integral bounds. In the preprint, we did not specify the integral bounds for the sake of readability, but we gladly comply with the editors request, and have added the integral bounds to eq. 1.

**EC:**  *My interpretation of the referee comment on the limitations of WEI and xWEI is the following: Extremeness in terms of a combination of duration and spatial extent is not necessarily equivalent to the relevance in terms of damage produced, and thus in terms of a measure that would likely be considered in the public. It could well be that an extreme event with high values of the xWEI would not produce any damage, and so the public might be surprised on the ranking. This is something that could be mentioned in the discussion.*

**AR:**  In our opinion, this issue is addressed concisely, yet prominently, in ll. 108-111 of the preprint:

*"The impact of such events on the ground, however, will vary dramatically based on governing hydrological and hydro-geomorphological processes as well as exposure and vulnerability. This is demonstrated by the*

*two extreme events that occurred in Lower Saxony, the impacts of which appeared to be less severe, or at least were less visible in the national media."*

In our view, this statement corresponds well to the editor's remark, so we'd prefer to not emphasize this point further, if the editor agrees.

**EC:** ***With respect to the referee's comment on duration, I am skeptic if the change to "duration level" would be helpful for the communication. In terms of Table 1, it may be worthwhile to state that after testing the different durations it turns out that the top event have a 24 or 48 hour duration. I am asking myself, however, if this outcome can be related to diurnal gauge readings entering the RADOLAN generation. This is clearly a question beyond the scope of your paper, but it could be interesting to examine this possibility.***

**AR:** We are not entirely sure whether we correctly understand the editor's comment. For the WEI-related analysis, we tested, for the sake of computational efficiency, only a fixed set of duration levels (1, 2, 3, 4, 6, 9, 12, 18, 24, 48, 72 h). Hence, the fact that the highest ranking events are most extreme at either 24 or 48 hours should not be overintepreted. For all five events, the actual duration at which the event was most extreme is somewhere between 24 and 48 hours which could be specified by testing durations at a higher resolution. We are sceptical that "diurnal gauge readings" (we assume the editor refers to totalizers that are read only at fixed times) have a large effect in this context.

In summary, we would prefer not to put too much emphasis on such a hypothesis, if the editor agrees. For clarification we added the following sentence to line 93:

> Note that due to computational efficiency only a fixed set of duration levels (1, 2, 3, 4, 6, 9, 12, 18, 24, 48 and 72 h) was considered. The actual duration at which the events were most extreme might be somewhere between these fixed levels.

**2. Response to referee 1**

**RC:** *[...] L3: firstly, the sentence "both rely....GEV parameters" might not be clear to everybody (I am putting myself in the shoes of someone not completely within the topic). Secondly, from how it is written it seems that the GEV distribution is the only one that can be used for the estimation of return periods of extreme events, which is not the case. I would suggest trying to make this sentence clearer and include some details about the link between return period and GEV parameters when you introduce the WEI in the main text.*

**AR:** We agree that this sentence should be clearer, and it is true that other data or methods could be used to estimate the required return periods. Yet, we should not go too much into the computational details within the abstract. We have tried to rephrase along these requirements.

> [...] Both require the estimation of return periods across different precipitation duration levels. For this purpose, previous studies determined annual precipitation maxima from radar composites in Germany, and estimated the parameters of a generalized extreme value distribution (GEV). Including the year 2021 in the estimation of GEV-parameters, [...]

**RC:** *L26: even if the authors claim that the WEI is increasingly used in the community (I guess they mean the meteorological one), for someone that is not within it the information reported about the WEI (229*

*log(yr)km) at this point of the paper is not straightforward to understand. The definition of WEI is indeed provided only at L60. I would therefore suggest trying to insert this number into a context and explain briefly what the extremity index is and how it is computed (or at least what variables are considered) such that a broader audience can have a feeling of what this number*

AR:   The unit of the WEI is indeed hard to communicate which is why we described the general concept underlying the WEI starting in line 22 of the preprint:

*"The WEI identifies the spatial and temporal scale at which an event was most extreme and allows to quantify the extremeness (rarity) of an event."*

To put the mentioned value of 229 log(yr)km into context the following sentence in the preprint states:

*"With this value, the event outranked all other events based on radar data that were previously classified in the period from 2001 to 2020."*

To further clarify this, we added a cross link to section 3 and the following sentence:

> The WEI identifies the spatial and temporal scale at which an event was most extreme and allows to quantify the extremeness (rarity) of an event by averaging return periods of all affected pixels and multiplying it with a measure of the spatial extent of this event (see Section 3).

We hope that this is an adequate compromise between summary and detail, given that the "Brief Communication" format provides limited space.

RC:   *L57-66: see my comments on L3.*

AR:   We replaced, in l. 63 of the preprint, the sentence "We use two different combinations of radar data and GEV parameter estimates" by the following statement in order to make clear that our choice to use DWD radar data together with the GEV distribution builds on previous research in that direction. However, we do not find it useful for the reader, at this point of the manuscript, to go into detail about alternative data or methods in order to retrieve the required return periods.

> Following Lengfeld et al. (2021), we used DWD's national radar composite data to first determine annual precipitation maxima, and then estimate the GEV parameters.

RC:   *L69-74: writing explicitly the formula through which the xWEI is computed would be helpful.*

AR:   Practically the xWEI is computed by an interpolation of a surface over Eta-curves Formally this corresponds to a double integral which can be expressed by the following formula, which we added to the revised manuscript:

$$xWEI = \int_{ln(t=1h)}^{ln(72h)} \int_{A=1km}^{(200km)} E_{tA}\, dA\, d(ln(t)) \qquad (1)$$

RC:   *Figures are not color-blind friendly, please consider change color scaled such that everyone can appreciate differences and color meanings.*

AR: We prepared Figure 1 and 2 with colorblind-friendly colormaps (see figures below), and used these in the revised manuscript. We used YlGnBu from ColorBrewer 2.0 (https://colorbrewer2.org/type=sequentialscheme=YlGnBun=9) for the colormap in Figure 1 and Tol_muted (https://zenodo.org/record/3381072.Y5MZ0iUxlhF) for the colors in Figure 2.

[Figure]

Figure 1: Left: Accumulated rainfall of the precipitation event from 12. July, 5:50 UTC to 15.July 2021, 5:50 UTC based on RADKLIM v2017.002 winterrath$_r adklim_2 018. Right$ : $Temporal development of the maximum hourly precipitation within the four districts Hagen, Mettmann, Cologne and Eusk$

RC: *Figures 2: do the gray lines represent the WEI of the July 2021 event? If yes, it should be specified somewhere. Moreover, I would rather representing it as a point (with different markers/colors depending on the RADKLIM product used to compute it).*

AR: Yes, the grey lines represent the WEI of the July 2021 event for the two different setups. We changed the lines to symbols and marked the WEI of the realtime setup with a star and of the climatological setup with an X in the new version of Figure 2 as shown in the figure below. We also added the information to the caption of Figure 2.

**3. Response to referee 2**

RC: *Uncertainty assessment:*
*I only found one major issue that needs to be addressed before publication, which is the lack of a proper uncertainty assessment. The authors could and should do more to quantify the uncertainty on the estimated*

[Figure]

Figure 2: Extremity calculated for 14 July 2021, 23:50 UTC, with operational RADOLAN data and GEV parameters obtained from RADKLIM 2001-2020 (realtime setup: solid lines, WEI marked with a star) and from reprocessed RADKLIM data (Version 2017.002) and GEV parameters from RADKLIM 2001-2021 (climatological setup: dashed lines, WEI marked with an X)

*return periods in the GEV, and how this uncertainty propagates to the WEI and xWEI. These are very important issues given the short available data record and the fact that the differences between the top 5 events aren't that large.*

AR:   We agree that a comprehensive uncertainty analysis of WEI and xWEI would be desirable. In a way this manuscript is also a statement regarding uncertainties as we show how WEI and xWEI values can change when extending the underlying time series.

A comprehensive uncertainty assessment however, as suggested by the referee, is beyond the scope of this study: apart from the quantification of the GEV parameter estimation uncertainty (which is computationally very expensive), the propagation and aggregation of these pixel-wise uncertainties in the computation of WEI and xWEI is anything but trivial, and would entail the development and validation of an adequate methodological approach. Such a development (and its documentation) exceeds the scope of a brief communication and would rather correspond to the format of a "research article". To this end, we would like to refer to the NHESS manuscript guidelines (https://www.natural-hazards-and-earth-system-sciences.net/about/manuscript_types.html):

*Brief communications are timely, peer-reviewed, and short (2–4 journal pages). These may be used to (a)*

*report new developments... ...disseminate information and data on topical events of significant scientific and/or social interest within the scope of the journal....*

Having said that, we provided information in Table 1 about the sensitivity of WEI and xWEI for the top 5 events when leaving the year of occurrence of the event out of the calculation of the GEV parameters as we already have done for July 2021 event. This allows for a better assessment of the uncertainty introduced by the short time series, which we believe is a major source of uncertainty in this study.

**RC:** *On the usefulness and need to rank extremes*
*I see value in studying extremes and their characteristics. However, I also wonder how useful it is to rank extremes over a given range of scales. Who needs such a ranking? And what can you really learn from a ranking that keeps changing over time as more data get available? Also, wouldn't such a ranking strongly depend on the lower/upper bounds for the calculation of xWEI?*
*Suggestion: add some discussion about the practical usefulness of ranking extremes and the scientific/practical limitations of the approach.*

**AR:** The ranking of events is the key to identify events that are exceptional with regard to a specific property (and hence "metric", in this case WEI or xWEI). Identifying such exceptional (high-ranking) events, in turn, is a relevant exercise for various reasons: to understand their occurrence and impacts, which are often fundamentally different from other (lower ranking) events, or to prioritise resources for event-based case or attribution studies. Furthermore the public often demands a "putting-into-context" of these events which can be communicated by ranks. The identification and characterisation of the most severe past events can help to adjust mitigation measures as these events can be used as benchmarks for further analysis.

For clarification, we changed the preprint version around line 106:

> Hence, the resulting rankings should not be over-interpreted, especially if we consider uncertainties regarding the radar rainfall estimation, the estimation of return periods and the selection of the spatial domain. ...... Nevertheless, ranks can help to compare events and serve as a tool to communicate an events extremeness to the public. Practically the identification and selection of these outstanding events can be used as benchmarks for further analysis and mitigation measures.

We agree with the referee that the choice of lower and upper bounds (in space and time, if we understand correctly) highly influences the values of the xWEI. For this study we used the same 200 x 200 km bounding box for all the different events, as we did in our previous paper (Voit and Heistermann, 2022), where we also discussed the choice of the spatial domain and considered durations in detail. While these choices are arbitrary to some extent, they are all the more important to make events comparable across space and time, and hence to allow for a ranking.

**RC:** *Alternative approaches*
*One limitation of WEI and xWEI is that they do not really tell us anything about how extreme an event was relative to others. Furthermore, the metrics involve the fitting of a GEV model, which comes with large uncertainty. Perhaps a different metric or different way of quantifying relative extremeness across scales should/could be considered?*
*Suggestion: add a few words about possible, alternative approaches to WEI and xWEI.*

**AR:** We are not sure whether we correctly understand the referee's statement that "one limitation of WEI and xWEI is that they do not really tell us anything about how extreme an event was relative to others." – as it is exactly what these indices were designed for. There are different concepts of quantifying the extremeness of heavy

precipitation events. "Simple" return periods or the exceedance of thresholds at gauges might be the most common ones, but do neither reflect the spatial extent of an event, nor give any information at which duration the event was most extreme. More advanced concepts like the Precipitation Severity Index (Caldas-Alvarez et al. 2022) consider the spatial extent, but also do not acccount for different durations explicitly. Another index was introduced by Ramos et al. (2017) which is based on precipitation anomalies and spatial extent of an event. None of these concepts considers and combines the co-occurrence of extremeness of an event at different duration levels, neither do these concepts identify the spatial and temporal scale on which an event was most extreme. So while we agree that WEI and xWEI are uncertain quantities, they have been shown to be useful measures for assessing and comparing the extremeness of heavy precipitation events - which is why we chose these measures in the context of this study. As this is a brief communication, we would prefer not to elaborate on alternative metrics in the introduction, since this would distract too much from the actual topic.

RC: *Temporal structure*
*Some information about the temporal structure of the July 2021 event would help the reader understand why this event was extreme over multiple scales, and how the water was distributed over time.*
*Suggestion: show a time series and/or give some information about peak rainfall rates, intermittency and standard deviation of rainfall rate over time for a fixed location. Fig1 covers the spatial aspect but there is no information about the time aspect so far.*

AR: We added barplots with temporal evolution of the hourly precipitation in four districts that were hit by the July 2021 event as shown in the figure below. The radar observations have a spatial resolution of 1 km x 1 km. We show the maximum hourly precipitation within each district to better understand temporal structure and added the following to the description of the event:

> The temporal development of the event (Fig. 1 right) indicates heavy precipitation on several temporal scales from showers of a few hours up to continuous rainfall for more than a day.

RC: *Stationarity assumption*
*There is an implicit stationarity assumption behind the whole study that should be mentioned.*
*Suggestion: Clearly mention the assumptions underlying your approach and the consequences they could have on the calculation of return values and (x)WEI. To reassure readers, I suggest you check whether there a trend in the precipitation extremes data over time. You can check this by fitting alternative GEV models with time-dependent shape or scale parameters and applying model selection based on likelihood ratio tests or AIC.*

AR: We now mention in the methods section of the revised manuscript that stationarity of the underlying GEV distribution is assumed when estimating the GEV parameters. The dataset of 21 years of data is not long enough for a reliable analysis of instationarity (even if we have to assume it exists). At the same time, the shortness of this times series makes stationarity a pragmatic and viable assumption. We comment on this issue in ll. 74-75 of the revised manuscript.

RC: *Equation 1*
*Please provide units for all quantities (A, T and E). What does the index i represent? The text does not say. Same for the index t. Please use ln() instead of log() to avoid ambiguity about the base of the logarithm.*

AR: $i$ represents a grid point of the radar composite, $t$ is used for a given duration in the original paper of Müller and Kaspar, where the extremity is defined. We agree it is more clear to use $ln$ instead of $log$. We changed this and explained $t$, $i$ and added the units.

[Figure]

Figure 3: Suggested revised version.

**RC:** *Table 1*
*Table 1: please provide units for WEI and xWEI*

AR: We added "ln(yr) km" as the unit for the WEI in Table 1. xWEi is dimensionless.

**RC:** *Table 1: I struggle to understand what you mean by "Duration". The caption says that the "Duration" is the timescale at which the maximum extremity was reached. But there are only two values (24h and 48h) for 5 events. I would have expected each event to have a peak at a different time scale. More generally, I think it would be useful to clarify what you consider to be an "event" and what the difference is between the "Duration" and the length of an "event". For example, is the average precipitation depth calculated at the event scale or over the duration indicated in the table?*

AR: We use the term "duration" in its standard meaning in hydrometeorology and extreme value statistics: the "duration" (or "duration level") is the length of a reference time interval over which the cumulative precipitation depth or the mean precipitation intensity is determined which could then be subjected to further (extreme value) analysis (as in intensity-duration-frequency curves, IDF). In this study, we analysed 11 different duration levels (1, 2, 3, 4, 6, 9, 12, 18, 24, 48 and 72 hours) in order to obtain the maximum corresponding return periods over the entire length of the event; and indeed, the 5 most extreme events according to the WEI all have the largest extremity at a duration level of 24 or 48 h. Again, for the computational details, we have to refer to the literature in terms of Lengfeld et al. (2021) and Voit and Heistermann (2022). In contrast to the referee's statement, we did not use the term "length of an event" in the manuscript, but in fact we need to define such a length for our analysis as it sets the time window in which we compute the return periods for the

abovementioned duration levels. While the length of an event could be considered as the time span between the first and the last raindrop, we admit that its definition involves some level of arbitrariness in determining the times of start and end. In any case, the length of an event could be shorter than an hour, but also longer than 72 hours, and is hence independent of the duration levels.

In summary, we suggest to use "duration level" instead of duration in the revised manuscript, in order to highlight the difference to "length of an event" (which is, however, not mentioned in the manuscript).

**RC:** *Table 1: it would be useful to indicate the change in WEI and xWEI for the other events as well. I understand that you are primarily interested in the changes for the July 2021 event. However, I also think that it's important to convey a general sense of how sensitive the WEI and xWEI metrics are to the inclusion/exclusion of particular year of data.*

AR: The WEI and xWEI for the other events did not change with the updated GEV parameters when including/excluding the year 2021 in the GEV parameter estimation. We now mention this in the manuscript. In addition, we computed the GEV parameters for the other four events without the year of their occurrence and added the WEI and xWEI for the events based on these parameters to get an idea of its sensitivity to the length of the time series (see also the comment by referee 3).

**RC:** *Min/Max bounds for integration*
*Section 3: For the calculation of WEI and xWEI, please clearly state the minimum/maximum bounds you took for integrating over the duration and area.*

AR: To keep the results comparable we used the same bounds for the computation of the xWEI as in our previous study (Voit and Heistermann, 2022): Duration levels from 1 h - 72 h and a bounding box of 200 km * 200 km around the centroid of the event. For the WEI the whole area affected by the event as defined in CatRaRE is taken into account for the calculation (Lengfeld et al., 2021).
We added this information in ll. 84 ff. in the revised manuscript.

**RC:** *l.87 The term "characteristic" duration was not properly defined.*

AR: We normally use the term "characteristic duration" for the durations at which an event has the highest extremity. In this brief communication we used the term in line 87, only. We replaced it by the following:

> This event had the largest extremity at a duration of 24 hrs...

**RC:** *l.103: I don't understand why the July 2021 could be considered a compound event. Please justify.*

AR: The referee is right that this point needs further explanation and clarification. Thieken et al. (2022) highlighted the flood events resulting from the August 2002 event in Saxony as a compound inland flood. During this event we observed extreme rainfall at rather short durations which caused flash floods as well as exceptional rainfall at longer durations (especially at the 48h duration) which caused fluvial floods and most likely led to an increased runoff coefficient due to the already saturated soils. We expect a similar behaviour for the July 2021 floods with different flood types most likely overlaying and amplifying each other, with the added complexity of hydrogeomorphic processes at various time scales (see Mohr et al. 2022 and Dietze et al. 2022). According to the definition below, as well as to the definition of the IPCC (Seneviratne et al. 2012), we are therefore inclined to describe these floods as "compound inland floods". However, we agree that the question whether the concept of a "compound event" could also apply to a precipitation event such as the one in July 2021 should be subject to further discussion within the scientific community. To clarify this point we changed the sentence in line 102 to:

> Thieken et al. (2022) framed the Elbe flood event that followed the August 2002 precipitation (rank 1 in Tab. 1) as a "compound inland flood", in the sense of different flood mechanisms overlaying and amplifying each other. Similar observations were made for the July 2021 event (e.g. Mohr et al. 2022). Given the exceptional xWEI values of both the August 2002 and the July 2021 events, it might be worth to discuss, in the future, whether heavy precipitation events that are extreme across scales could be framed as hydro-meteorological compound events.

**RC:** *According to Leonard et al. (2014), "A compound event is an extreme impact that depends on multiple statistically dependent variables or events". According to Zhang et al. (2021), compound extremes are defined as 1) two or more extreme events occurring simultaneously or successively, 2) combinations of extreme events with underlying conditions that amplify the impact and 3) a combination of events that are not extreme individually but lead to an extreme event or impact when combined.*
*In the case of the July 2021 event, I do not see why this event should be labeled as "compound". It just appears to have been extreme over multiple spatial and temporal scales at the same time. Please elaborate!*

 AR:  We refer to our previous answer.

**4.  Response to referee 3**

**RC:** *[...] The two WEI values are not obvious in Fig. 2. Moreover, they are labeled as "realtime" and "climatological" which is not the main factor of the difference as authors state at l. 66-67.*

 AR:  We highlighted the two WEI values with two different symbols instead of the grey lines (see new figure below) to make the difference clearer. The reviewer is right, the difference is mainly caused by including/excluding the year 2021 in the estimation of the GEV parameters. For the sake of simplicity, and to improve the readability of the manuscript, we had chosen the terms "realtime" and "climatological" for the two different scenarios as defined in lines 63-66, and hence use these names for the labels in Figure 2. To emphasize the difference between the two setups again in the figure, we also mention it in the figure caption and changed it as follows:

> Extremity calculated for 14 July 2021, 23:50 UTC, with operational RADOLAN data and GEV parameters obtained from RADKLIM 2001-2020 (realtime setup: solid lines, without including the year 2021 in the GEV parameter estimation, WEI marked with a star) and from reprocessed RADKLIM data (Version 2017.002) and GEV parameters from RADKLIM 2001-2021 (climatological setup: including the year 2021 in the GEV parameter estimation, dashed lines, WEI marked with an X).

**RC:** *Significant decrease in both WEI and xWEI values when including the year 2021 proofs that the presented estimation of return periods is not very robust. It is natural because of the rather short length of the data series. Nevertheless, because this is the main concern of the communication, a short sensitivity analysis could be done, e.g. by the "leave-one-out" technique.*

 AR:  We agree that an analysis in terms of the suggested leave-one-out technique would be a useful extension in order to assess the sensitivity of the indices to the fact whether an event is included in the GEV parameter estimation or not. For the revised manuscript, we calculated WEI and xWEI for the other four events in Tab. 1 using GEV parameters computed *without* the years of their occurrence, and added this information to Table 1.

[Figure]

**RC:** *The unique impacts of July 2021 were not only because of the precipitation event itself but partly also due to the enhanced soil moisture at its beginning. In my opinion, this fact should be more stressed in the discussion.*

**AR:** We agree with the reviewer that the antecedent soil moisture might have had an effect on runoff formation. We would prefer, however, to be a bit cautious with regard to that effect: first, the antecedent wetness is not directly related to the event-based extremity indices of precipitation; second, the role of antecedent wetness with regard to runoff formation and hence the impacts of the events is still subject to scientific discussion. On that basis, we added the following statement to the conclusions section, after l. 103 of the original manuscript:

> Mohr et al. (2022) also suggested that, originating from the weeks before the event, the antecedent soil moisture in the affected region was relatively high, favoring the occurrence of saturation excess and thus an enhanced rainfall-runoff transformation.

**RC:** *line 80: adding the word "even" into the sentence ("are even less pronounced") would make it more clear in my opinion;*

**AR:** Thanks for the comment. We added the word "even" to the sentence.

**RC:** *line 84: I suggest to add "respectively" at the end of the sentence.*

**AR:** We added "respectively" at the end of the sentence.

**References**

Caldas-Alvarez, A., Feldmann, H., Lucio-Eceiza, E., and Pinto, J. G.: Scale-dependency of extreme precipitation processes in regional climate simulations of the greater Alpine region, Weather Clim. Dynam. Discuss. [preprint], https://doi.org/10.5194/wcd-2022-11, in review, 2022.

Dietze, M., Bell, R., Ozturk, U., Cook, K. L., Andermann, C., Beer, A. R., Damm, B., Lucia, A., Fauer, F. S., Nissen, K. M., Sieg, T., and Thieken, A. H.: More than heavy rain turning into fast-flowing water – a landscape perspective on the 2021 Eifel floods, Nat. Hazards Earth Syst. Sci., 22, 1845–1856, https://doi.org/10.5194/nhess-22-1845-2022, 2022.

Lengfeld, K., Walawender, E., Winterrath, T. and Becker, A.: CatRaRE: A Catalogue of radar-based heavy rainfall events in Germany derived from 20 years of data, Met. Zet., doi:10.1127/metz/2021/1088, 2021

Leonard, M., Westra, S., Phatak, A., Lambert, M., van den Hurk, B., McInnes, K., et al. (2014a). A Compound Event Framework for Understanding Extreme Impacts. Wires Clim. Change 5, 113–128. doi:10.1002/wcc.252

Mohr, S., Ehret, U., Kunz, M., Ludwig, P., Caldas-Alvarez, A., Daniell, J. E., Ehmele, F., Feldmann, H., Franca, M. J., Gattke, C., Hundhausen, M., Knippertz, P., Küpfer, K., Mühr, B., Pinto, J. G., Quinting, J., Schäfer, A. M., Scheibel, M., Seidel, F., and Wisotzky, C.: A multi-disciplinary analysis of the exceptional flood event of July 2021 in central Europe – Part 1: Event description and analysis, Nat. Hazards Earth Syst. Sci., 23, 525–551, https://doi.org/10.5194/nhess-23-525-2023, 2023.

Müller, M. and Kaspar, M.: Event-adjusted evaluation of weather and climate extremes, Natural Hazards and Earth System Sciences, 14, 473–483, 2014.

Ramos, A. M., Trigo, R. M. and Liberato, M.L.R. (2017). Ranking of multi-day extreme precipitation events over the Iberian Peninsula, International Journal of Climatology, 37(2), doi:/10.1002/joc.4726

Seneviratne, Sonia and Nicholls, Neville and Easterling, David and Goodess, Clare and Kanae, Shinjiro and Kossin, James and Luo, Yali and Marengo, Jose and McInnes, Kathleen and Rahimi, Mohammad and Reichenstein, Markus and Sorteberg, Asgeir and Vera, Caroline and Zhang, Xuebin: Changes in climate extremes and their impacts on the natural physical environment, A Special Report of Working Groups I and II of the Intergovernmental Panel on Climate Change (IPCC), DOI: 10.7916/d8-6nbt-s431, 2012.

Thieken, A. H., Samprogna Mohor, G., Kreibich, H., and Müller, M.: Compound inland flood events: different pathways, different impacts and different coping options, Nat. Hazards Earth Syst. Sci., 22, 165–185, https://doi.org/10.5194/nhess-22-165-2022, 2022.

Voit, P. and Heistermann, M (2022). Quantifying the extremeness of precipitation across scales, Natural Hazards and Earth System Sciences, doi:10.5194/nhess-22-2791-2022

Zhang W, Luo M, Gao S, Chen W, Hari V and Khouakhi A (2021). Compound Hydrometeorological Extremes: Drivers, Mechanisms and Methods. Front. Earth Sci. 9:673495. doi: 10.3389/feart.2021.673495